# Experimental *Mycobacterium microti* Infection in Bank Voles (*Myodes glareolus*)

**DOI:** 10.3390/microorganisms10010135

**Published:** 2022-01-10

**Authors:** Enric Vidal, Judit Burgaya, Lorraine Michelet, Claudia Arrieta-Villegas, Guillermo Cantero, Krystel de Cruz, Jennifer Tambosco, Michelle Di Bari, Romolo Nonno, Maria Laura Boschiroli, Bernat Pérez de Val

**Affiliations:** 1IRTA, Centre de Recerca en Sanitat Animal (CReSA, IRTA-UAB), Campus UAB, 08193 Bellaterra, Spain; enric.vidal@irta.cat (E.V.); judit.burgaya@gmail.com (J.B.); claudia.arrieta@irta.cat (C.A.-V.); guillermo.cantero@irta.cat (G.C.); 2Animal Health Laboratory, National Reference Laboratory for Tuberculosis, Paris-Est University, Anses, 94700 Maisons-Alfort, France; Lorraine.michelet@anses.fr (L.M.); krystel.decruz@anses.fr (K.d.C.); Jennifer.tambosco@anses.fr (J.T.); maria-laura.boschiroli@anses.fr (M.L.B.); 3Department of Food Safety, Nutrition and Veterinary Public Health, Istituto Superiore di Sanità, 00161 Rome, Italy; michele.dibari@iss.it (M.D.B.); romolo.nonno@iss.it (R.N.)

**Keywords:** voles, wild rodents, tuberculosis, *Mycobacterium microti*, experimental infection

## Abstract

Voles are maintenance hosts of *Mycobacterium microti*. In line with the goal to eradicate tuberculosis (TB) in livestock, the role of this mycobacteria needs to be assessed since it might interfere with current *M. bovis*/*M. caprae* surveillance strategies. To better understand the pathogenesis of TB in voles, an experimental infection model was set up to reproduce *M. microti* infection in laboratory Bank voles (*Myodes glareolus*). Two infection routes (intragastric and intraperitoneal) and doses (10^5^ and 10^6^ CFU/0.1 mL) were assessed. Voles were culled at different post-infection time points. Serology, histopathology, acid-fast bacilli staining, qPCR, and mycobacterial culture from tissues were performed. In addition, qPCR from feces and oral swabs were conducted to assess bacterial shedding. The model allowed us to faithfully reproduce the disease phenotype described in free-ranging voles and characterize the pathogenesis of the infection. Most animals showed multifocal and diffuse granulomatous lesions in the liver and spleen, respectively. Less frequently, granulomas were observed in lungs, lymph nodes, muscles, and salivary gland. Mycobacterial DNA was detected in feces from a few animals but not in oral swabs. However, one contact uninfected vole seroconverted and showed incipient TB compatible lesions, suggesting horizontal transmission between voles.

## 1. Introduction

*Mycobacterium microti*, a member of the *Mycobacterium tuberculosis* complex (MTBC), is the main causative agent of tuberculosis (TB) in wild rodents, particularly field voles (*Microtus agrestis*), bank voles (*Myodes glareolus*), wood mice (*Apodemus sylvaticus*), yellow-necked mice (*Apodemus flavicollis*), and shrews (*Sorex araneus*) [1,2,3,4,5]. Cases of TB due to *M. microti* have also been described in humans [6], a wide-range of domestic species, including cattle [7,8], goats [9], pigs [10], dogs [11], cats [12,13], and wildlife other than micromammals, such as meerkats [14], squirrel monkeys [15], badgers [16], foxes [17], deer [18], and wild boar [19,20], presenting a distribution across different European countries, including France, UK, Italy, Spain, Germany, Austria, and Switzerland [3,9,18,19,20,21].

Although differentiation of *M. microti* from the other members of MTBC is possible mainly by genotyping methods [21], it is rarely diagnosed because of the apparently lower susceptibility in humans and livestock [6,7,9], but also due to *M. microti*’s slow growing rate that that impedes its isolation by classical bacteriological methods [22]. However, the extended distribution of this mycobacteria, together with the newly sensitive hosts presents a concern for the identification and management of its role in the control strategies of animal TB. Indeed, it has been suggested that it may interfere with current *M. bovis*/*M. caprae* diagnosis conducted in livestock in the frame of TB eradication campaigns [8]. This poses a major concern in low bovine TB prevalence settings, such as in the Pyrenees, where *M. microti* has recently been detected in different hosts [20,23].

Since voles are described as the main maintenance host for this *M. microti*, an experimental infection was designed to reproduce the natural disease and better understand the pathogenesis and transmission of TB in voles.

## 2. Materials and Methods

### 2.1. Experimental Animals and Housing

Bank voles (*Myodes glareolus*) were bred at the facilities of the *Istituto Superiore di Sanità* (Rome, Italy). Then, experimental voles, aged between 80 and 167 days, were transported to the BLS-3 facilities of IRTA-CReSA (Bellaterra, Spain) and were kept in a controlled environment at a room temperature of 22 °C, 12 h light–darkness cycle and 60% relative humidity in HEPA filtered cages (both air inflow and extraction) in ventilated racks. The voles were fed ad libitum, observed daily, and their clinical status assessed once a week. Periodically, voles were weighed and oral swabs and individual fecal samples were obtained. Males and females were housed separately and a maximum of 4 animals per cage. All experiments took place at the Biocontainment Level 3 animal facility at IRTA-CReSA.

### 2.2. Study Design and Experimental Infection

Two separate experiments were conducted sequentially.

#### 2.2.1. Experiment 1

For the first experiment, a *M. microti* field strain used as inoculum was initially isolated in a dog of the Ariege region (France). The strain was selected on the basis of its spoligotype profile, SB0423 (Mbovis.org), which has been detected in different mammals in cross-border regions of France [23] and Spain [20] (Ariege, Haute Garonne, and Catalonia).

The inoculum was prepared at Anses (Maisons-Alfort, France) as follows: The isolate was subcultured in Middlebrook 7H9 medium with mycobactin and a titer of 3 × 10^7^ CFU/mL was determined by measurement of optic density at the beginning of the stationary phase. Once the master seed of *M. microti* was titered, it was diluted in sterile PBS at final suspension of 10^6^ CFU/mL.

This experiment involved two groups (*n* = 8, male and female balanced, thus, a total of 16 animals) each group with a different inoculation route: (1) intragastric (IG), 0.1 mL of 10^5^ CFU/0.1 mL of *M. microti* using a plastic flexible intragastric mice probe and a 1 mL syringe; and (2) intraperitoneal (IP) via inoculation of 0.1 mL of 10^6^ CFU/mL of *M. microti* with a 25 G needle and a 1 mL syringe. The inoculation was carried out under general gaseous anesthesia (Isofluorane 5%). Two time points were established at 28- and 114-days post inoculation (DPI) for euthanasia and postmortem examination. Animals that died spontaneously or had to be euthanized for humanitarian reason were also analyzed.

#### 2.2.2. Experiment 2

For the second experiment, the inoculum was prepared as following: A *M. microti* SB0423 isolate, obtained from the spleen of a vole of the experiment 1 (IP group, euthanized at 28 dpi), was subcultured in Middlebook 7H9 medium (BD Diagnostics, Sparks, NV, USA) for 28 days at 37 ± 1 °C. This second inoculum was titrated by qPCR as following: an aliquot of 1 mL of the suspension was inactivated at 75 °C for 1 h. In parallel, an aliquot of the *M. microti* master seed (3 × 10^7^ CFU/mL) used for the challenge inoculum of the experiment 1, was also inactivated and then ten-fold serially diluted to establish titrated standards. DNA samples were extracted and amplified using the ID Gene™ spin universal extraction and ID Gene™ *Mycobacterium tuberculosis* complex Duplex commercial kits (ID.vet, Grabels, France), respectively, according to the manufacturer procedures. Amplification was performed in a 7500 fast real-time PCR system (Applied Biosystems, Walham, MA, USA). *M. microti* CFU genomic equivalents were calculated by the extrapolation of Ct value obtained for each DNA sample according to previously described [24]. Once titrated, the remaining subcultured suspension was diluted in sterile PBS at a final suspension of 10^7^ CFU/mL.

In this experiment, the IP route was chosen and a higher dose was used, it consisted of a group (*n* = 14, male and female balanced) inoculated IP with 0.1 mL of 10^7^ CFU/mL of *M. microti*. Four additional unchallenged animals were housed in 4 cages, each of them sharing cage with 3 inoculated animals to assess the occurrence of horizontal transmission. An early time point was established at 28–37 dpi (a first batch of 3 animals were euthanized at 29 dpi and were grouped with other three animals that were found dead at 28 and 37 dpi, due to TB-unrelated causes) and a second late time point at 69 dpi for euthanasia and postmortem examination. Two additional voles remained uninoculated and unexposed to infected animals and were used as negative controls.

### 2.3. Ethics Statement

All experimental animal procedures were approved by the Animal Research Ethics Commission of the *Generalitat de Catalunya* (procedure number FUE-2020-01337124 and ID 46KZF9ZM4), according to current European legislation for the protection of experimental animals (86/609/EEC, 91/628/EEC, 92/65/EEC, and 90/425/EEC).

### 2.4. Postmortem Examination and Pathological Evaluation

Animals were humanely euthanized upon reaching end point criteria or at established time points by general gaseous anesthesia (Isofluorane 5%) and subsequent decapitation. After visual examination of the viscera, samples of liver, spleen, and lung tissue were obtained and frozen at −75 °C, the remaining tissues were immediately immersed in formalin. Two weeks later the tissues were routinely processed for histopathology, i.e., paraffin embedded, 3–4 µm sections were obtained and further stained with Hematoxylin and Eosin (HE) for morphological assessment. Ziehl Neelsen (ZN) stain was used to detect AFB. The tissues processed for microscopic evaluation included: kidney, liver, lung, thoracic lymph node (LN), spleen, stomach, intestines, mesenteric LN, cervical LN, inguinal LN, and any tissue with visible lesions.

To objectively assess the progression of the TB lesions, a pathology index was established based on a mixed scoring system that integrated different parameters evaluated separately: the absence or presence of lesions (0/1) inguinal LN, cervical LN, and peritoneum; for the spleen a semiquantitative scoring system (0–3) was established (from 0—absence of lesion— to 3—extensive splenomegaly due to diffuse granulomatous inflammatory infiltrate—scores 1, 2, and 3 were assigned to mild, moderate, and evident levels of the same lesional pattern, respectively); thoracic lesions were evaluated as summatory of the absence/presence (0/1) of TB lesions in the lung and thoracic lymph nodes combined. Finally, lesions in the hepatic tissue were quantitatively evaluated using Image J software [25] including different parameters: number of granulomas found in five 10× randomly selected liver fields, the area of the granulomas was calculated, as well as the absence/presence (0/1) of a central area on necrosis and absence/presence (0/1) of mineralization. For the pathology index, the liver score included three parameters: necrosis (0/1), Mineralization (0/1) and a third one regarding the size of granulomas. For this third parameter a score of 0 was given when the animal’s total liver granuloma area median was below (smaller granulomas) the global liver granuloma area median (5729.985 μm^2^) and 1 when it was above (bigger granulomas).

### 2.5. Mycobacterial Cultures

Liver, spleen, and lung tissues were thawed and sliced with sterile scissors and subsequently mechanically homogenized in 1.5 mL of sterile distilled water, and an aliquot of 1 mL of each homogenate was separated for subsequent bacterial load assessment by qPCR. The remaining 0.5 mL was decontaminated with 0.5 mL of oxalic acid at 5% *w*/*v* for 30 min and then neutralized with 250 μL of NaOH 1 M. Afterwards, samples were centrifuged at 2451× *g* for 30 min. Supernatants were discarded and pellets were suspended in 1 mL of sterile PBS. Suspensions were cultured as following: 100 μL were inoculated in BBL tubes and incubated in BACTEC MGIT 320 system (BD diagnostics, Sparks, MD, USA), 100 μL were cultured in Middlebrook 7H11 plates (BD diagnostics, Sparks, MD, USA), and a swab was immersed in the remaining suspension for culture in Löwenstein–Jensen with pyruvate and Coletsos solid medium tubes (BD Diagnostics, Sparks, MD, USA). Growth in positive cultures were confirmed as MTBC by multiplex PCR [26]. A culture was considered negative when no growth was observed in Middlebrook 7H11 plates (at 28 days), BACTEC MGIT (at 42 days) or solid medium tubes (at 90 days).

### 2.6. Mycobacterial Load Assessment by qPCR

Aliquots of 1 mL of liver, spleen, and lung homogenates of each vole were inactivated at 75 °C for 1 h. The *M. microti* master seed was used to generate the qPCR standard curve. DNA samples from homogenates and standards were extracted and amplified, and *M. microti* CFU genomic equivalents were calculated as described above.

### 2.7. Mycobacterial Excretion Assessment by qPCR

Oral swabs and feces were collected from each at animal at days: 7, 14, 21, 28, and 56 p.i. (experiment 1), and 13, 41, and 69 p.i. (experiment 2). Oral swabs were immersed and cut in microtubes containing 1 mL of PBS, while feces was suspended and homogenized in 1 mL of PBS. All samples were inactivated at 100 °C for 10 min at stored at −20 °C.

DNA was extracted from oral swabs using an LSI MagVetTM Universal Isolation Kit (Life Technologies, Villebon sur Yvette, France) with a KingFisherTM Flex automate (ThermoFisher Scientific, Villebon sur Yvette, France), following the manufacturer’s instructions. DNA was extracted from feces with the FastDNA™ Spin Kit for Soil (MP Biomedicals, Eschwege, Germany). Real-time PCR were carried out in a 25 µL reaction mix containing TaqMan™ Fast Advanced Master Mix (ThermoFisher Scientific, Villebon sur Yvette, France), 300 nM forward and reverse primers, 250 nM probes, sterile water, and 5 µL of DNA template. Thermocycling conditions were 50 °C for 2 min (1 cycle), followed by one cycle of 20 s at 95 °C and 40 cycles of 3 s at 95 °C and 30 s at 60 °C. PCR inhibition was tested (Diagenode, Seraing, Belgium). Positive detection of *M. microti* was established on the basis of a positive response for IS1081 and IS6110 (*Mycobacterium tuberculosis* complex) [27].

### 2.8. Serology

Blood was obtained upon sacrifice through decapitation. Sera were separated and a MTBC IgG ELISA was carried out and interpreted as previously described [28]. Briefly, sera were analyzed for antibodies against the MTBC-specific MPB83 antigen (Lionex, Braunschweig, Germany) by a homemade IgG indirect ELISA. A sample was classified as positive (i.e., seroconversion) when ΔOD (sample wells Optical Density—OD—at 450 nm minus the blank well OD at 450 nm) ≥0.2.

### 2.9. Data Analysis

A randomized design was used for data analyses. Differences in the spleen pathology score, liver granuloma area, overall pathology index, and bacterial loads in the spleen (log_10_ CFU equivalents) between the early and late infection groups were compared using the one-tailed Mann–Whitney non-parametric test. Correlation between the spleen pathology score and bacterial loads in the spleen was performed by a non-parametric Spearman rank test. The statistical analysis and visualizations were carried out using the R software version 4.1.1, and the following packages: dplyr (1.0.7), tidyverse (1.3.1), ggplot2 (3.3.5), ggpubr (0.4.0), hrbrthemes (0.8.0), viridis (0.6.1), viridisLite (0.4.0), heatmaply (1.2.1), and plotly (4.9.4.1).

## 3. Results

### 3.1. Comparison of Infection Routes

In experiment 1, at 28 dpi no macroscopic or microscopic lesions were observed in neither IP nor the IG groups. In the IP group 2/4 voles seroconverted and 3/4 yielded low titer positive PCR and culture results from the spleen samples. No indication of TB infection was found in any of the IG animals euthanized at 28 dpi (see Table 1 for a summary of experiment 1 results and Appendix A for the complete data set).

At late time points, the four animals of the IP group showed positive qPCR in the spleen with similar or higher titers than the inoculum (from ~5 × 10^4^ to ~10^7^ CFU equivalents, see Appendix A), whereas the IG group showed 3 out of 4 positive spleens, one of them with a lower titer than the inoculum (~8 × 10^2^ CFU equivalents).

An animal of the IP group was found dead at 58 dpi (the cause of death could not be determined) and it already showed TB compatible lesions. The TB lesions in this animal were mild and were not considered to be the cause of death. The remaining animals of the IP group, sacrificed at 114 dpi when the end point of the study was established, showed evident macroscopic and microscopic TB compatible lesions.

Macroscopically a moderate increase in the spleen size (splenomegaly) was observed only in the two animals of the IP group sacrificed at 114 dpi. Microscopically small granulomas were observed in the liver with a miliary distribution, composed of macrophages, the occasional multinucleated giant cell, and small foci of mineralization. The splenic parenchyma was invaded by a diffuse granulomatous infiltrate with scant multinucleated giant cells and multifocal foci of mineralization (Figure 1). The same infiltrate could be observed in cervical, thoracic, mesenteric, and inguinal lymph nodes. In one animal a focal area of granulomatous infiltrate could be observed in the salivary gland. Finally, and probably because of the inoculation procedure, a subcutaneous granulomatous infiltrate was observed in the left inguinal region as well as foci of granulomatous infiltrate in the peritoneum, in contact with pancreatic tissue. No lesions were observed in the lung or the kidneys. A moderate to low amount of AFB was observed in all the lesions.

The results obtained with the IG inoculated group at later time points were inconsistent: one animal was found dead at 95 dpi with lesions similar to those described above for the IP group at 114 dpi, including splenomegaly, but it also showed a visible subcutaneous granuloma, consisting of a histiocytic proliferation infiltrating the skeletal muscle of the left anterior limb. Ziehl Neelsen staining evidenced that most of the macrophages were heavily loaded with abundant AFB (Figure 2). This animal also showed the highest bacterial load of the experiment in the spleen, liver, and lungs (~4 × 10^8^, ~3 × 10^5^ and ~3 × 10^7^ CFU equivalents calculated by qPCR, respectively). Regarding the two remaining animals, euthanized at the end point of the experiment (114 dpi), one showed no lesions (yet a positive culture result in the liver) and the other had a lesional pattern equivalent to that observed in the IP group, including granulomatous miliary hepatitis, diffuse granulomatous splenitis and thoracic and inguinal granulomatous lymphadenitis.

MTBC DNA was not detected from oral swabs by PCR in any animal, even in that with the granulomatous sialadenitis. Regarding the feces, in the first experiment only one animal showed double positivity to both IS*6110* and IS*1081* targeted qPCRs (criterion to consider a sample as positive). This result was obtained in one animal (BV-25) of the intragastric group at 7 dpi, probably corresponding to the remains of the inoculum since the feces of this animal were negative at later time points (Appendix A).

### 3.2. Comparison of IP Dosages

For the experiment 2 the IP infectious route, which had demonstrated a more consistent result, was selected as well as an inoculum with a higher titer (10^6^ CFU/0.1 mL) to try to optimize the model and shorten the disease onset (See Table 2 for a summary of experiment 2 results and Appendix A for the complete data set).

All the voles euthanized at the early time point (28–37 dpi) seroconverted and developed TB compatible pathology comparable or slightly more severe than the one observed in 114 dpi animals of the first experiment. Additionally, 2/5 voles also presented pulmonary granulomas and 3/5 showed granulomatous infiltrate in the peritoneum (Table 2, Appendix A). PCR revealed *M. microti* titers in the spleen equivalent to, or slightly higher, than those contained in the inoculum (Table 2).

All the animals euthanized at a late time point in experiment 2 (69 dpi) showed a more severe and advanced pathology, also characterized by miliary granulomatous hepatitis, but in this case some of the granulomas (in 3/8 animals) were larger and had a central area of necrosis. Diffuse granulomatous splenitis was also observed fully obliterating the splenic parenchyma. Granulomatous lymphadenitis could be observed in all studied lymph nodes (thoracic, cervical, mesenteric, and inguinal) but only 2/8 voles had granulomatous lesions in the lung. As with experiment 1 all IP inoculated animals showed some degree of subcutaneous inguinal granulomatous infiltrate as well as peritoneal granulomatous infiltrates, the latter were observed not only in the vicinity of pancreatic parenchyma but also surrounding the kidneys, without infiltrating the capsule nor the renal parenchyma.

### 3.3. Disease Progression

To study the progression of the disease several parameters were semiquantitatively and quantitatively evaluated in early and late time points in the animals from Experiment 2 (Figure 3, Figure 4 and Figure 5).

The pathology scoring system applied for the spleen (from ‘0’ normal spleen to ‘3’ severe granulomatous splenomegaly, Figure 3A) revealed a statistically significant increase in the late time point (*p* = 0.021, Figure 3B). The average score for the early time point was 1 and for the late time point was 1.875. The late time point also showed an increased bacterial load measured in this tissue by the qPCR although not statistically significant (*p* = 0.091, Figure 3C). Accordingly, pathology score and bacterial load showed a statistically significant direct correlation (Spearman *ρ* = 0.7, *p* = 0.008, Figure 3D).

Neither the number nor the average area of liver granulomas in five 10× random fields showed statistically significant differences between time points. However, qualitative differences were observed: in the late time point, higher number of larger granulomas were found (Figure 4), and additionally 3/8 animals had granulomas with central necrosis that were not observed in any of the early time point animals. Mineralization was observed in both time points indistinctly and none of the granulomas presented capsulation. At the late time point, the animals presented a higher absolute number of granulomas compared to the early time point (62% increase). This increase is mainly explained by a larger number of small granulomas, which is compatible with a recent generation.

A pathology index summarizing the whole pathology, was assessed. The index was calculated adding the spleen score, a liver score (from 0 to 3, which considered the granuloma area as well as the presence/absence of calcification and central necrosis), a thoracic score (from 0 to 2, i.e., presence/absence of lesions in lung and/or thoracic LN), and the absence/presence (0–1) of TB lesions in peritoneum, inguinal, and cervical lymph nodes, Figure 5A). The score was significantly higher in the late time point (*p* = 0.01, Figure 5B). This could be explained largely by the higher spleen scores due to more severe splenomegaly, presence of TB lesions in the cervical lymph node (only observed in the late time point) and the presence of central necrosis in the liver granulomas, also particular to the late time point. The number of hepatic granulomas did not significantly increase over time, neither did the proportion of animals with pulmonary lesions, which remained unchanged.

All fecal samples were negative at early time points while MTBC DNA was detected in five animals at 69 dpi by IS*6110* or IS*1081* qPCRs, although only one of them (BV-40) was considered positive (detected by the two qPCRs), thus suggesting fecal excretion of *M. microti* at the late time point (Appendix A). In this second experiment, no positive PCR results were obtained from the oral swabs either.

### 3.4. Horizontal Transmission

Only a few inoculated animals showed positive or inconclusive qPCR results pointing to fecal *M. microti* shedding, whereas no oral excretion was detected. The only histological evidence of potential mycobacterial excretion was the presence of lung granulomas found in four animals of the second experiment (two in the early and two in the late group) and one focus of granulomatous sialadenitis in one animal of the first experiment (114 dpi, IP).

However, one out of the three contact animals that survived to the end of the experiment seroconverted. *M. microti* was not detected in the spleen, lungs, and liver neither by q PCR nor culture. Upon histological examination a single multinucleated giant cell was observed in the inguinal lymph node and, after evaluating five Ziehl–Neelsen-stained serial sections of each organ of this animal, one acid fast bacillus compatible image was observed in the cervical lymph node (Figure 6).

## 4. Discussion

TB in wild voles was described in 1934 by Wells and Oxon [5], who initially referred to it as ‘the vole acid-fast bacillus’ [29]. The aim of the present study was to set up an experimental model of *M. microti* TB in one of its natural reservoirs to better characterize the infection and pathogenesis. A specific goal was to precisely determine the phenotype caused in Bank Voles by the *M. microti* strain identified circulating in the Pyrenees in other mammal species [20,23]. Both inoculation routes tested successfully reproduced a TB infection in bank voles with lesions resembling those previously reported in free-ranging voles [3]. The most common lesions described in field voles, consisting in granulomatous hepatitis, splenitis, and lymphadenitis, were observed in most of the infected animals in our study. Another type of typically reported lesion are skin granulomas [2,3], which we have not observed in this experimental model, this is most likely due to the inoculation route chosen since skin lesions are hypothesized to be the entry portal through bites from other infected animals. We have observed, indeed, the presence of subcutaneous granulomas in the inoculation point of the intraperitoneal group. The miliary lesional pattern observed in the liver parenchyma and the presence of diffuse granulomatous infiltrate in the spleen, in both IG and IP routes, strongly suggest that a hematogenous spread of mycobacteria took place in these animals. Lesions in other viscera such as the lungs, skeletal muscle, or the salivary gland were also observed but with a much lower frequency. No differences in the proportion of animals with these lesions were observed in the high dose vs. the low dose experiment, suggesting that individual host-related variation factors, rather than the dose change, influence the spread of the disease. Given that some animals presented lung and salivary gland lesions suggests that shedding of mycobacteria is likely to occur.

The oral intake of mycobacteria is a likely infectious route in the field, from the environment or even through consumption of infected carcasses. One of the IG-inoculated animals showed, indeed, muscular lesions similar to those described in field voles [3]. However, the results obtained in the IG route were less consistent than the IP route, which generated a 100% attack rate even with the lower dose at the early time point. The titers obtained by qPCR in the early time point low dose IP inoculated spleens were, in fact, lower than the titer of the inoculum, and thus these results might be interpreted as residual inoculum and not a true infection, but considering the evolution of all the remaining IP animals towards a more severe pathology and titers similar or higher than those inoculated, we interpret these results as early stages of infection. Another potential artifact of the IP route is the presence of granulomatous infiltrate and mycobacteria in the peritoneum of the animals, observed in the pancreatic parenchyma and perirenally, probably resulting from the direct deposition of mycobacteria in the peritoneal cavity. An alternative explanation for these lesions would be the hematogenous spread of mycobacteria from lesions in other viscera to the serous membranes, in a mechanism similar to that causing TB pearly disease in other species [30]. In any case, the IP route was the chosen inoculation route for the second experiment and the one we recommend for future experimental bioassays.

A log increase in the inoculum titer was tested for the second experiment focused on the IP route. This increase in the dose approximately accelerated the observed phenotype twice. The lesions observed in the early time point (28–37 dpi) in the IP high dose animals were equivalent or even more severe that those observed at 114 dpi in the lower dose group. In any case, longer incubation times would likely lead to the appearance of more lesions in the lungs, salivary glands and probably the kidneys. An increased dose, however, allowed us to optimize the model shortening the incubation period needed to observe widespread lesions.

TB is usually a slowly progressive disease. The rapid progression of the lesional phenotype observed in our experiments can be explained by the relatively high doses inoculated. Indeed, one log increase in the dose resulted in a noticeable acceleration of the phenotype. In the field voles are likely to be exposed to variable doses, probably much lower than the ones used here. Descriptions of field studies report a proportion of animals with positive cultures of *M. microti* in the absence of visible TB lesions [3], this suggests that the severity of lesions observed is heterogeneous and that even animals with latent *M. microti* infection might exist. This is in agreement with the variable results observed in the first experiment with intragastric exposure, which likely mimics one of the natural exposure routes. In a laboratory setting, however, the aim of an experimental model is to reproduce a given phenotype in a controlled manner; thus, those conditions giving a fast and reproducible phenotype are the ones to be met. Even if the result is not representative of the full spectrum of the disease phenotype in the field. Eventually, a need to design control tools for wild micromammal populations with high TB prevalence will benefit form a well-characterized model.

In the second experiment, we assessed the pathological and bacteriological findings at two time points after *M. microti* experimental infection to characterize the progression of the disease in the bank vole model. As expected, the severity of lesions increased in target tissues at 70 dpi compared to 30 dpi (i.e., higher mean pathology score in the spleen and number of large granulomas in the liver). Mycobacterial loads in the spleen also tended to increase at late time points. The presence of lesions or mycobacterial isolation in other localizations also confirmed the progression of the infection. Regarding the liver lesions, not only qualitative changes indicative of pathology progression where observed (presence necrosis and increased granuloma size), but also an increase in the total number of granulomas observed in the late time point animals (although not statistically significant) that could be explained by the formation of new granulomas due to a continued hematogenous spread of mycobacteria. This fact could explain the lack of significant differences when comparing both time points as regards to the size of granulomas.

Moreover, mycobacterial shedding results suggested that infected animals could also be infectious at the late time point. Even though mycobacterial DNA was not detected from oral swabs in any case throughout both experiments, mycobacterial DNA in feces was detected in a few animals, although with a very low bacterial burden. Therefore, a fecal–oral transmission cannot be ruled out. Little evidence could be gathered from the histological study where only four animals with lung lesions and one with salivary gland lesions could be observed that could support mycobacterial excretion through sputum (both from lower respiratory tract exudates and/or from saliva). The characteristic skin lesions described in field TB cases in voles could be explained by transmission trough bites from infected individuals excreting mycobacteria through the saliva [3]. Still, one of the contact-uninoculated animals seroconverted. Interestingly, this animal shared cage with two inoculated animals in which lesions were not observed neither in the lungs nor in the salivary gland, and additionally qPCR and culture form their lungs, as well as from the feces and oral swabs of these animals that were also negative. Therefore, there is no evidence of *M. microti* transmission between these animals. However, certain evidence of infection was found in the seropositive contact animal, consisting of a single multinucleated giant cell and a single AFB compatible structure in the studied sections (Figure 6), that precluded a solid confirmation of horizontal transmission in our experimental setting. Further studies with more prolonged incubation times are needed to assess this. An alternative explanation for seroconversion could be an oral exposure to mycobacteria from the inoculation point of its cage mates at day 0.

## 5. Conclusions

*M. microti* IP and IG challenge successfully reproduced a TB phenotype resembling field infection in bank voles.

The IP route provides more consistent results than IG (100% attack rate and earlier pathology onset). Increasing two logs the infectious titer inoculated accelerates (approximately times two) the onset of TB pathology, thus optimizing the model.

Longer studies are required to confirm likely horizontal transmission between voles.

As disease progresses, splenomegaly increases, the number of liver granulomas remains unchanged but granulomas become bigger and develop a central necrotic core. For surveillance matters in field voles, microscopic examination of the liver is recommended in the absence of macroscopically visible lesions, particularly in voles with a certain degree of splenomegaly. Spread to other organs, such as the lung or salivary glands, leading to mycobacterial shedding, probably depends on host-related individual factors.

## Figures and Tables

**Figure 1 microorganisms-10-00135-f001:**
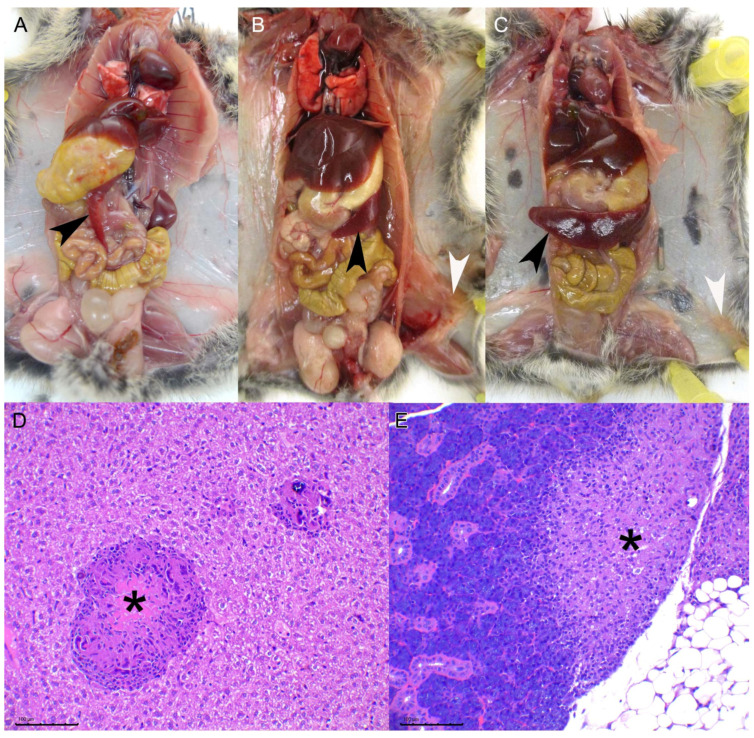
TB lesions in intraperitoneally inoculated bank voles: (**A**): Mock inoculated bank vole necropsy. Notice the thin, almost transparent, spleen (black arrowhead) (BV2). (**B**): Bank vole euthanized 29 dpi (BV15, 10^6^ CFU dose), notice a moderate increase in spleen size (black arrowhead) and the presence of a granulomatous subcutaneous lesion in the inguinal region (white arrowhead). (**C**): Necropsy of a bank vole euthanized at 69 dpi (BV40, 10^6^ CFU dose). Notice the severe splenomegaly (black arrowhead) and also a small inguinal subcutaneous granuloma (white arrowhead). (**D**): Liver of BV16 (10^6^ CFU dose, 69 dpi) showing larger granuloma with a central area of necrosis (asterisk). HE, bar 100 µm. (**E**): Salivary gland of BV8 (10^4^ CFU dose, 114 dpi). Notice the granulomatous infiltration of the glandular parenchyma (asterisk) as well as the surrounding adipose tissue (on the right). HE, bar 100 µm.

**Figure 2 microorganisms-10-00135-f002:**
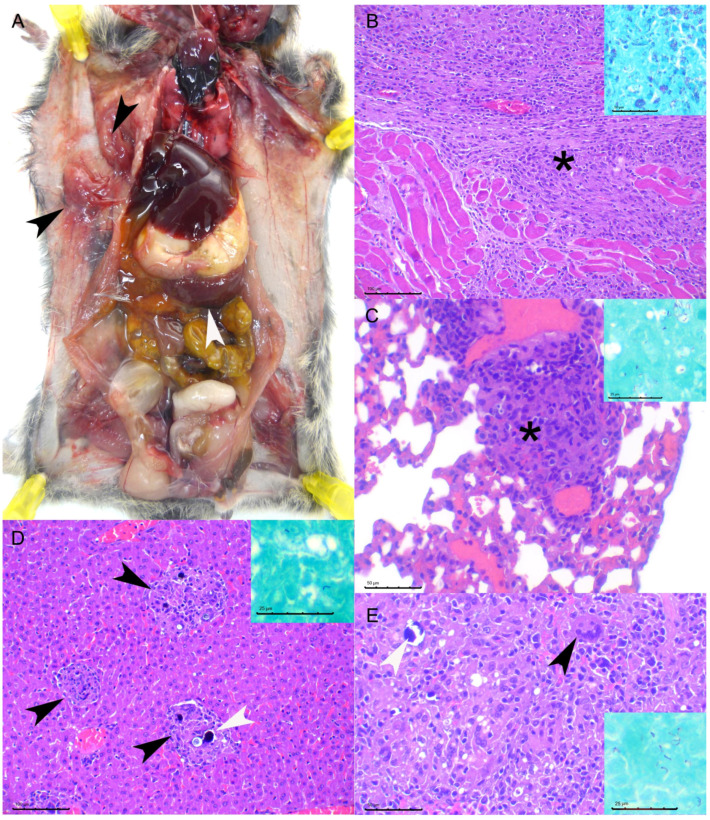
TB lesions in an intragastric inoculated bank vole (BV20, 95 DPI): (**A**): Macroscopically an evident increase in size of the spleen (white arrowhead) was observed, along with a subcutaneous granuloma in the axillar region (black arrowheads). (**B**): Microscopically the axillar lesion corresponded to a diffuse granulomatous inflammatory proliferation (asterisk) that infiltrated muscle fibers. Remains of skeletal muscle fibers can be observed on the bottom half of the image. Hematoxylin and Eosin staining (HE), bar 100 µm. Inset: most macrophages were heavily loaded with abundant acid-fast bacilli (AFB). Ziehl–Neelsen staining (ZN), bar 25 µm. (**C**): A single small granuloma could be observed in the lung (asterisk). HE, Bar 50 µm. Inset: abundant AFB were observed in this lesion. ZN, bar 25 µm. (**D**): Multifocal small granulomas were observed in the liver (black arrowheads) some of which showed punctiform mineralization foci (white arrowhead). HE, bar 100 µm. Inset: a few AFB were observed in these lesions. ZN, bar 25 µm. (**E**): A diffuse granulomatous infiltrate was observed in the spleen with multinucleated giant cells (black arrowhead) and focal mineralization foci (white arrowhead). HE, Bar 50 µm. Inset: abundant AFB were observed in this lesion. ZN, bar 25 µm.

**Figure 3 microorganisms-10-00135-f003:**
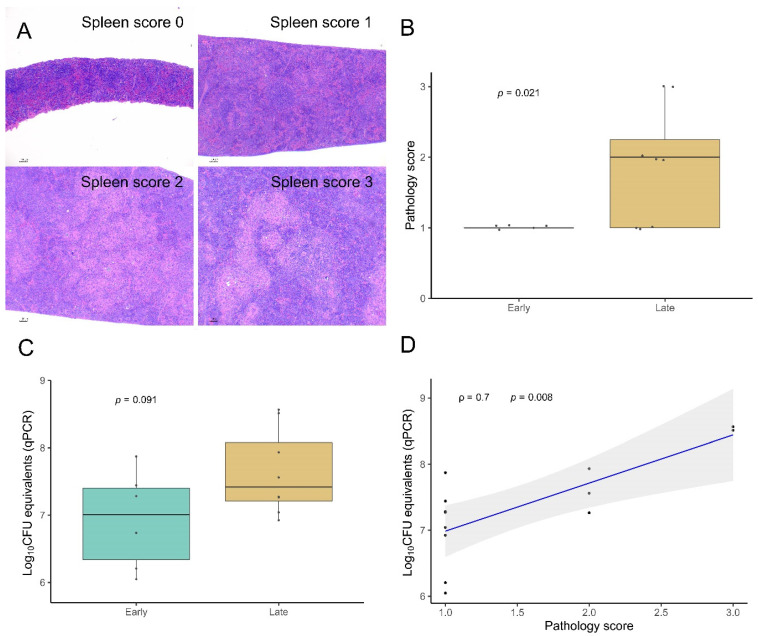
Splenic lesions progression: (**A**): Spleen pathology scoring. From 0—absence of lesion to 3—extensive splenomegaly. Scores 1, 2, and 3 correspond to mild, moderate, and evident levels of the same lesion pattern. Scale bars 100µm. (**B**): Spleen pathology score difference between the early and late time points. Significances determined by one-tailed Mann–Whitney test. (**C**): Differences in the bacterial load between the early and late time points as log_10_ CFU equivalents measured by qPCR. Significances determined by one-tailed Mann–Whitney test. (**D**): Correlation determined by between the bacterial load (log_10_ CFU equivalents) and the spleen pathology score. Spearman rank test, *ρ*: Spearman rho.

**Figure 4 microorganisms-10-00135-f004:**
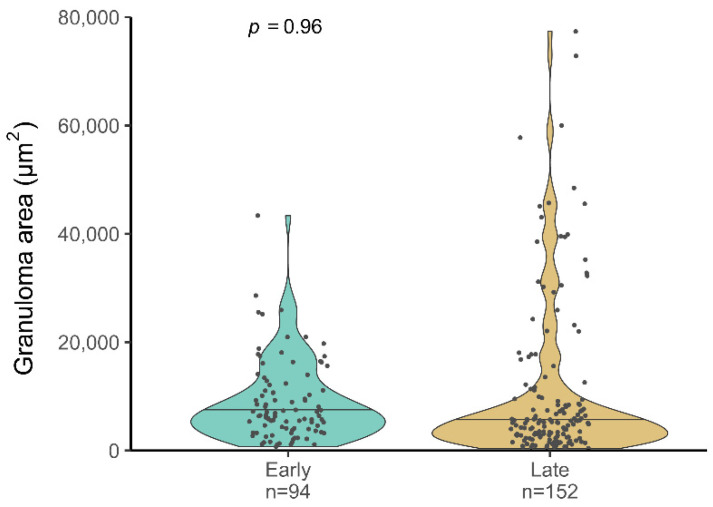
Liver pathology measured as the granuloma area. The median granuloma area (μm^2^) at the early and late stages is represented by horizontal lines. Number of granulomas: N = 94 at early stage and N = 152, at a late stage. Significances determined by one-tailed Mann–Whitney test.

**Figure 5 microorganisms-10-00135-f005:**
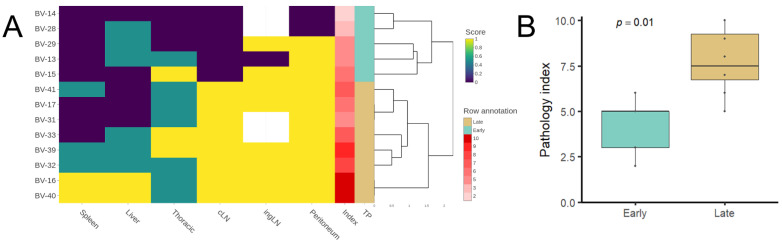
Pathology progression summary. (**A**): Heatmap of normalized scores for each bank vole. The rows correspond to the ids (BV-XX), and the columns to the score given to each organ according to the disease progression and/or the presence/absence of lesions. Spleen: 0–3 according to the spleen score. Liver: 0–3 according to the presence/absence of calcification, central necrosis, and the granuloma size (>total media = 1, <total media = 0). Thoracic: 0–2, presence/absence of lesions in the lung and thoracic lymph node. *cLN* (cervical lymph node), *ingLN* (inguinal lymph node), and peritoneum: 0–1 according to presence/absence of lesions. The row annotation corresponds to the pathology index assessed, corresponding to the sum of the different scores, and to the Early/Late time points (TP). (**B**): Pathology index at the Early and Late stages. Significances determined by one-tailed Mann–Whitney test.

**Figure 6 microorganisms-10-00135-f006:**
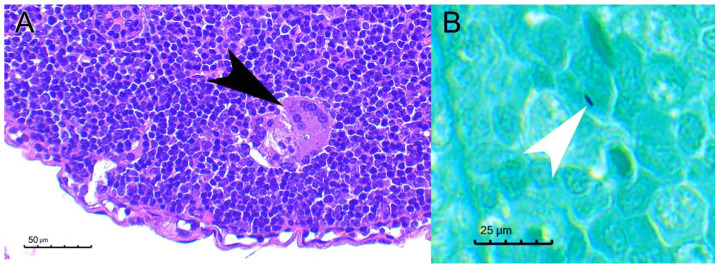
Pathological findings in a non-challenged contact bank vole. (**A**): A single multinucleated giant cell (black arrowhead) was observed in the inguinal lymph node of BV18 (contact uninoculated animal). HE, bar 50 µm. (**B**): After examining several serial sections a single acid-fast bacilli (white arrowhead) could be observed in the cervical lymph node. ZN, bar 25 µm.

**Table 1 microorganisms-10-00135-t001:** Summary of the results for Experiment 1. The numbers indicate the number of positive results/total animals per group (or available samples). dpi: days post inoculation.

Experiment 1	Timepoint	Serology	MTBC qPCR			MTBC Culture			TB Compatible Lesions		
Route			Spleen	Liver	Lung	Spleen	Liver	Lung	Spleen	Liver	Lung
Intragastric (10^5^ CFU)	Early (23–28 dpi)	0/4	0/4	0/4	0/4	0/4	0/4	0/4	0/4	0/4	0/4
	Late (58–114 dpi)	2/3	3/4	1/4	2/4	2/4	3/4	1/4	2/3	2/4	0/4
Intraperitoneal (10^5^ CFU)	Early (17–28 dpi)	2/4	3/4	0/4	0/4	3/4	0/4	0/4	0/4	0/4	0/4
	Late (95–114 dpi)	2/2	4/4	2/4	2/4	2/4	2/4	1/4	2/3	3/3	0/3

**Table 2 microorganisms-10-00135-t002:** Summary of the results for Experiment 2. The numbers indicate positive results/total animals per group (or available samples). dpi: days post inoculation.

Experiment 2	Timepoint	Serology	MTBC qPCR			*M. microti* Culture			TB Compatible Lesions		
Route			Spleen	Liver	Lung	Spleen	Liver	Lung	Spleen	Liver	Lung
Intraperitoneal (10^6^ CFU)	Early (28–37 dpi)	3/3	6/6	5/6	6/6	4/6	3/6	2/6	4/5	6/6	2/6
	Late (69 dpi)	8/8	8/8	5/8	4/8	8/8	7/8	0/8	8/8	8/8	2/8
Contact	Late (69 dpi)	1/4	0/4	0/4	0/4	0/4	0/4	0/4	0/4	0/4	0/4

## Data Availability

The raw data used in the present study are available in Appendix A. Code availability https://github.com/jburgaya/article_microti. Last accessed on 17 December 2021.

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
