# Peer review of "Experimental Mycobacterium microti Infection in Bank Voles (Myodes glareolus)"

_microorganisms, 2022, doi:10.3390/microorganisms10010135_

Round 1
Reviewer 1 Report
This manuscript describes the development of an experimental M. microti infection model in bank voles. The study design is good and the paper is well written. The data is comprehensively analysed and the results are convincing.
My main comments on the paper relate to clarification of particular issues that would improve the manuscript.
Lines 51-55: This is very confusing and needs to be re-phrased (the syntax and grammar is not quite correct). Consider breaking it up into two or more sentences.
Line 72 (section 2.2.1 title): Should be something like ‘M. microti inoculum preparation’ or ‘M. microti inoculation’
Line 73: Should be ‘were conducted sequentially’.
Lines 93 – 115 (Experiment 2). I was very confused reading this section. There seems to be two separate subsections here, if I understand it correctly? The first is heat inactivation of M. microti and DNA extraction to generate qPCR standards. The second relates to the experimental infections. If this is correct then they are currently mixed up and it is difficult to follow.
Please consider revising along the following lines: Bring the experimental infection protocol (starting at line 107, ending at 115) up to the beginning of this section, as this is the description of the Experiment 2. Then follow up with the heat inactivation and DNA extraction as a separate sub-section (or separate paragraph) clearly pointing out what the purpose of this sub-experiment is?
Line 99: ‘tittered’ has a completely different to what is intended! Use ‘titered’ instead.
Line 108: M. microti in italics.
Line 245: Use a colon (:) after ‘inconsistent’ instead of a full stop.
Line 371: Delete ‘for the first time’.
Line 383: Should be ‘miliary’
Lines 393: ‘carcasses’ belonging to what (range of) species?
Two other general comments:
The inoculation doses employed for the experimental infections are very high and unlikely to be encountered in nature by bank voles. It is likely therefore that the accelerated pattern of progression of infection is likely to reproduce the pathogenesis equivalent to lower natural doses, but still at the high end and sufficient to generate lesions. As in many wild species the pathogenesis may be subject to a dose response. It is not implausible to suggest that the average natural exposure dose of bank voles is much lower than the doses that give rise to severe pathology (as seen here). Is there any evidence that bank voles may harbour latent M. microti with a different profile of pathogenesis? This merits a mention in the discussion highlighting that the equivalent pathogenesis seen in the wild and in the high dose experimental infection may be representative of a biased observation as a consequence of the dose response, and that there may be a different pathogenesis in voles infected naturally with much lower doses.
It is stated in lines 57-58 that the purpose of the study is to understand pathogenesis and transmission of M. microti in voles. How do you intend to proceed with these new observations? Is there a practical application for knowing this new information? In other species, it might be used to evaluate vaccines etc, I can't see how this would be practical for voles? Perhaps include ‘the vision’ at the end of the Discussion section.
=
Reviewer 2 Report
Please see the attached document for general and specific comments regarding this manuscript.
Review for Microorganisms
Manuscript ID: microorganisms-1493276
Title: Experimental Mycobacterium microti infection in bank voles (Myodes glareolus)
Authors: Enric Vidal , Judit Burgaya , Lorraine Michelet , Claudia Arrieta-Villegas , Guillermo Cantero , Krystel De Cruz , Jennifer Tambosco , Michele Angelo Di Bari , Romolo Nonno , Maria Laura Boschiroli , Bernat Pérez de Val *
Abstract: Voles are maintenance hosts of Mycobacterium microti. In line with the goal to eradicate tuberculosis (TB) in livestock, the role of this mycobacteria needs to be assessed since it might interfere with current M. bovis / M. caprae surveillance strategies. To better understand the pathogenesis of TB in voles, an experimental infection model was set up to reproduce M. microti infection in laboratory Bank voles (Myodes glareolus). Two infection routes (intragastric and intraperitoneal) and doses (105 and 106 CFU / 0.1 mL) were assessed. Voles were culled at different post-infection time points. Serology, histopathology, acid-fast bacilli staining, qPCR and mycobacterial culture from tissues were performed. In addition, qPCR from faeces and oral swabs were conducted to assess bacterial shedding. The model allowed us to faithfully reproduce the disease phenotype described in free ranging voles and characterize the pathogenesis of the infection. Most animals showed multifocal and diffuse granulomatous lesions in the liver and spleen, respectively. Less frequently, granulomas were observed in lungs, lymph nodes, muscles, and salivary gland. Mycobacterial DNA was detected in faeces from a few animals but not in oral swabs. However, one contact uninfected vole seroconverted and showed incipient TB compatible lesions, suggesting horizontal transmission between voles
General comments: This paper develops a model for tuberculosis in bank voles which are a natural wildlife host of M. microti. The paper does a good job of exploring two common routes of laboratory challenge and determining which route better suits the laboratory while representing faithfully what is seen in wild animals. The findings in this paper are valuable contribution to the study of a zoonotic tuberculous bacterium. Photographs and histopathologic images are of excellent quality.
Specific comments:
Line 43: needs to say microti’s…
Line 79: followingàfollows
Line 84 & line 108 : clarify, “This experiment involved two groups (n = 8, male and female balanced)” and “(n = 14, male and femal balanced) … Additional (n = 4) unchallenged animals were housed in 4 cages along with 3 109 inoculated animals to assess the occurrence of horizontal transmission.”
Line 141: why just five randomly chosen fields vs the entire liver section?
Line 144-146: This would benefit from rewording or being broken into 2 sentences. I believe this is trying to differentiate “larger vs smaller” granulomas but it took several reads to understand.
Line 170: feràwere
Double checking, fecal samples from individuals NOT bulk cage samples were used?
Line 203: “In experiment 1, at 28 dpi no macroscopic or microscopic lesions were observed in neither IP nor the IG groups, but 2/4 IP inoculated voles seroconverted and 3/4 yielded low titer positive PCR and culture results from the spleen samples.”
- Suggested rewriting for clarity à “In experiment 1, at 28 dpi no macroscopic or microscopic lesions were observed in neither IP nor the IG groups. In the IP group, 2/4 voles seroconverted and 3/4 yielded low titer positive PCR and culture results from the spleen samples.”
Line 217: The remaining animals, sacrificed at 114 dpi when the end point of the study was established, showed evident macroscopic and microscopic TB compatible lesions. Macroscopically a moderate increase of the spleen size (splenomegaly) was observed 220 only in the two animals sacrificed at 114 dpi.”
- So were macroscopic lesions visible in all of the remaining 114 dpi animals or did some only show microscopic lesions? If the latter, then the sentence should read, “The remaining animals, sacrificed at 114 dpi when the end point of the study was established, showed evident macroscopic and/or microscopic TB compatible lesions.”
Figure 1 & 2: these are mislabeled. Figure 1 should actually be figure 2 and vice versa. In the text, figure 1 is supposed to depict IP inoculated animals, but the provided figure 1 clearly titles is at intragastric. The same incongruity is found with the text of figure 2 and the actual figure 2.
Line 314: following the statement that the size and number of granulomas was not statistically significant, you go on to mention non-significant findings, which are interesting trends. Please reiterate that while observed, these were non-significant to avoid confusion.
Line 426: suggest next to, “an increase in the total number of granulomas observed in the late time point (although non-significant) could be explained…
